# Single-Center Experience with Simultaneous Mural Aortic Thrombosis and Peripheral Obstructive Disease in Pre-COVID-19 and COVID-19 Era

**DOI:** 10.3390/diagnostics13061208

**Published:** 2023-03-22

**Authors:** Filippo Benedetto, Francesco La Corte, Domenico Spinelli, Gabriele Piffaretti, Santi Trimarchi, Giovanni De Caridi

**Affiliations:** 1Department of Biomedical Sciences and of Morphological and Functional Image, University of Messina, 98100 Messina, Italy; 2Department of Medicine and Surgery, School of Medicine, Varese University Hospital, University of Insubria, 21100 Varese, Italy; 3Department of Clinical Sciences and Community Health, University of Milano, 20122 Milan, Italy

**Keywords:** aortic thrombosis, peripheral artery disease, follow-up, COVID-19

## Abstract

Background: Mural aortic thrombosis associated with chronic peripheral obstruction of the lower limbs is an unusual event. Repeated embolism of instability aortic mural thrombosis caused acute limb ischemia (Rutherford 2 classification) in patients with peripheral arterial disease (PAD). We report a single-center experience for patients with transmural aortic thrombosis and peripheral artery disease. Methods: We retrospectively analyzed data of 54 patients with aortic mural thrombus disease with PAD presentation, treated at our center between 2013 and 2022. Results: Thirty patients (six with proven SARS-CoV-2 infection) underwent hybrid or staged treatment for an aortic lesion and for lower limb ischemia, by the placement of an endovascular aortic stent graft and a femoro-distal or a popliteal-distal bypass graft. The remaining 24 cases were only subjected to an intravascular treatment of the thoracic or abdominal aorta. Transient renal failure occurred in three patients. No embolic events were detected during the procedures. Aortic-related mortality was reported in just one patient who died from multiple organ failure. There was an embolic stroke in one patient with proven SARS-CoV-2 infection, three major amputations in patients with proven SARS-CoV-2 infection and no aortic-related mortality. Conclusions: Stent coverage of complex aortic lesions, alone or in association with a distal bypass graft, supports this approach in a variety of settings. The COVID-19 pandemic caused an increased mortality and amputation rate.

## 1. Introduction

A distal arterial embolism is a relatively common problem that carries increased morbidity and mortality. The amputation rate following acute limb ischemia is estimated at 13–14%, while mortality is at 9–12% [1]. The most common source of peripheral and visceral embolization is represented by thrombi in the left side of heart. The most frequent non-cardiac causes include aortic pathologies such as aneurysmal lesions, dissections, penetrating ulcers or traumatic lesions [2,3]. Aortic mural thrombus in a non-aneurysmal minimally atherosclerotic or normal aorta is a rare clinical entity and an unusual cause of peripheral arterial embolization [4]. Oliver. et al. published the first described case of thromboembolism from the thoracic aorta in 1967 [5]. Primary aortic mural thrombus has been classified by Verma et al. into four types according to the anatomical lesion location: type I in the ascending thoracic aorta and arch (a and b), type II (a and b) in the descending thoracic aorta, type III in the visceral abdominal aorta and type IV in the infrarenal aorta. The optimal management of these patients is still controversial and it depends on the thrombus location and morphology, the symptoms and the patient’s general conditions [4]. Treatment includes systemic anticoagulation [6] and open surgical or endovascular procedures [6,7]. Surgical treatment has been rather restrictively used, mainly because operative morbidity is still considered comparatively high [6]. The advancement of endovascular techniques and devices provides a minimally invasive alternative to patients otherwise deemed poor surgical candidates [8]. Recurrent peripheral embolism as the initial presentation of an underlying occult aortic mural thrombus is seldom diagnosed, and it could be the cause of peripheral artery disease (PAD), which necessitates lower limb revascularization. Coagulation disorders are common in COVID-19 and are associated with disease severity. With the incidence of a viral disease following inflammatory responses, an imbalance in procoagulant and anticoagulant mechanisms occurs, with endothelial dysfunction playing a major role [3]. COVID-19 patients with a history of coagulation problems cause concern about presenting more risks than other patients [4]. This study report our experience with stent coverage of complex aortic lesions, which was associated with a distal bypass graft in patients with repeated embolism of unstable aortic-mural-thrombosis-caused acute limb ischemia (Rutherford class IIb classification) and PAD.

## 2. Materials and Methods

### 2.1. Study Design and Setting

We retrospectively analyzed clinical and imaging data from 54 consecutive patients diagnosed with presence of an aortic embolic source, after being hospitalized at our center between January 2013 and June 2022 for chronic peripheral arterial disease. This study was approved by Ethical Review Board of our institution (number 372 bis/22), and written informed consent was obtained from all patients.

### 2.2. Variables

In all patients, the peripheral symptoms, with different severities, represented the initial clinical manifestation associated with echo color doppler finding of recent thrombosis of arterial segments of lower limbs or the presence of patent tibial femoral arteries, with no plantar arch. All patients underwent cardiac evaluation with transthoracic and transesophageal echocardiography. All patients underwent CT angiography (CTA), while the same examination was performed after revascularization of lower limb and further recurrent peripheral embolism when a cardiac source was excluded. All CTA images were acquired on multilayer CT scanners with 16 or 64 detector configuration. Non-contrast images were obtained at first, followed by acquisition of 1 mm axial images from the top of aortic arch to femoral arteries after intravenous contrast injection. The acquired CTA data were transferred to an OsiriX 3.9 workstation (Pixmeo SARL, Bernex, Switzerland) for analysis. For thoracic and descending abdominal aorta, aneurysmal dilation was defined as an aorta having a diameter of at least twice that patient’s normal contiguous aortic caliber. The presence and the location of a mural thrombus was also evaluated.

Of these 54 patients, 39 had CLI and all were clinically classified by Rutherford classification system for PAD. Among these, 8 patients with proven SARS-CoV-2 infection were clinically classified by Rutherford classification system for acute limb ischemia in class IIb. All patients were evaluated by preoperative duplex ultrasound or intraoperative digital subtraction angiography (DSA). Data collection included patient demographics, risk factors for vascular disease, symptoms leading to diagnosis, relevant comorbidities, management, and diagnostic and therapeutic outcomes. Cardiovascular risk factors were considered as follows: chronic heart failure (defined as LVEF < 30% or biatrial dilatation), COPD (FEV1 < 50%; B2+; O2-chronic therapy), CKD (serum creatinine > 2 g/dl; GFR < 60 mL/min), CVD (prior CEA, CAS, or stroke). Treatment complications and follow-up outpatient assessments were reviewed. All imaging studies were evaluated. All patients underwent endovascular stent placement for aortic disease, while some of them also underwent open peripheral revascularization procedure. Depending on timing, some patients underwent synchronous peripheral revascularization (when endovascular and open procedures were simultaneously performed in a single operating session) or in stages, in a dedicated surgical room with a mobile fluoroscopic C-arm (Philips BV Pulsera, Philips Medical Systems, Eindhoven, Netherlands and Eurocolombus Alien E, Eurocolumbus srl, Milan, Italy). The anesthetic strategy was chosen according to surgical needs and surgical risk of patients. Completion check was performed by duplex continuous wave ultrasound examination after peripheral revascularization procedures and digital subtraction angiography after aortic procedures. **Decision making for open vs. endovascular procedures based for significant patients’ comorbidity.** All patients underwent postoperative antithrombotic and antiplatelet therapy (LMWH and acetylsalicylic acid).

### 2.3. Follow-Up Protocol

Follow-up included duplex ultrasound checks for lower extremity revascularization procedures at 1, 3, 6 and 12 months and annually thereafter. Aortic follow-up was performed with CT angiography at one month, then duplex ultrasound with contrast medium and/or echo colors and CT angiography were alternated every six months for abdominal aorta, while a CT scan angiography was performed every year for thoracic aorta. Primary patency was defined according to Rutherford reporting standards [9].

### 2.4. Statistical Analysis

Statistical analysis was performed using R (R Core Team, 2019; R: A language and environment for statistics computing. R Foundation for Statistical Computing, Vienna, Austria. https://www.r-project.org/), Continuous variables were reported as median with interquartile range (IQR) or as mean ± standard deviation (SD) as appropriate. Categorical variables were described with counts and percentages. Survival and patency were estimated using Kaplan–Meier method. Patients were divided into subgroups based on procedures’ timing, and the subgroups were compared using Log-Rank test. Statistical tests were considered significant when *p*-value was <0.05.

## 3. Results

Among the 54 patients (43 male, 11 female), 30 underwent hybrid endovascular and surgical treatment for mural aortic thrombosis and PAD possibly induced, while 24 underwent only the aortic endovascular procedure. The average age was 68.8 ± 10.2 years. Eight patients had contracted a previously proved SARS-COV-2 infection (PSC2I); based on COVID’S onset timing, one patient of eight submitted to hybrid procedures had a confirmed SARS-CoV-2 infection 1 month before the intervention, three patients from 2 to 3 months before the intervention, two patients from 5 to 6 months before the intervention, one patient 8 months before the intervention and the last patient 10 months before the intervention. The median duration of follow-up was 28.5 months (IQR 41.3–13.3). Three patients were lost to follow-up. Risk factors for vascular disease were hypertension (83%), diabetes (38%), smoking (44%), renal failure (22%), COPD (37%) and coronary heart disease (37%). In total, 79% were already taking antihypertensive drugs and 55% were taking antiplatelet drugs (Table 1). CTA scan revealed unstable thoracic (19 cases) or abdominal (35 cases) aortic thrombus of patients who had symptoms of chronic peripheral arterial disease with limb-threatening ischemia (CLTI) (35 cases) or typical distal embolism of “junk kick” type (19 cases) (Figure 1). All CLTI patients were classified according to the Rutherford system as: class 4, 15 patients; class 5, 13 patients; and class 6, 11 patients. In nine patients with cli (Rutherford’s 4), only intravascular aorta treatment solved the symptoms. Twelve patients underwent one-step procedures, while 22 patients underwent hybrid one-step procedures: the median delay between the two interventions was 26 days (IQR 35.5–11). Among the patients with PSC2I, six underwent hybrid endovascular and surgical treatment for mural aortic thrombosis and PAD (Figure 1 and Figure 2). Two patients with PSC2I underwent endovascular procedures.

In 19 cases, coverage of the thoracic aorta embolic source was performed, while in 35 cases, the abdominal aorta was treated. The thoracic aorta was covered for an average of 133.5 mm (median 150 mm, IQR 150–101 mm). Four patients underwent femoral bypass, while eight patients underwent femoral-popliteal bypass, 17 underwent femoro-tibial bypass and 1 underwent popliteal bypass. In 23 cases, they were performed using the large saphenous, while a prosthetic graft was used in 7 cases. A total of 11 patients underwent an aortic procedure after peripheral revascularization. All patients underwent postoperative antithrombotic and antiplatelet therapy (LMWH and acetylsalicylic acid), which was then discharged with antiplatelet therapy. Twenty-four patients underwent just the aortic procedure of embolic source coverage and they were therefore discharged with a prescription for antiplatelet therapy (acetylsalicylic acid 100 mg per day) in combination with cilostazol for the treatment of PAD. The operational details are summarized in Table 2 and Table 3. No embolic events were detected during procedures. Transient renal failure occurred in three patients (7%) and myocardial infarction occurred in one patient (2%), while there were no cerebrovascular events and no cases of spinal cord ischemia. In two cases, we observed a recurrent embolism after aortic coverage. Aortic-related mortality was reported in only one case of a patient who died of multiple organ failure. Overall survival was 98% at 30 days, 95% at 6, 12 and 24 months, and 77% at 48 months. The overall primary patency of distal bypass grafts was 91% at 30 days and 87% at 6, 12 and 24 months. Based on the timing of peripheral revascularization, four patient subgroups were identified: in cases of synchronous aortic and peripheral procedures, primary patency was 100% during the entire follow-up, while in cases of phased procedures, it was 87% at 30 days and 79% at 6, 12 and 24 months; in particular: in cases of aortic surgery before peripheral revascularization, primary patency was 93% at 30 days and later, while in cases of peripheral revascularization before the aortic procedure, primary patency was 87% at 30 days and 75% at 6, 12 and 24 months. We did not record any differences in terms of these results, which were related to the aortic district of the embolic source or to the type of endografts that were used.

In patients with PSC2I, the average age was 63.4 ± 9.2 years. The median duration of follow-up was 11.2 months (IQR 17.3–5.8). Risk factors for vascular disease were hypertension (81%), diabetes (42%), smoking (61%), renal failure (26%), COPD (40%), coronary heart disease (29%) and corticosteroid therapy (65%); 73% were already taking antihypertensive drugs and 57% were taking antiplatelet drugs. Table 1 A CTA scan revealed unstable thoracic (two cases) or abdominal (six cases) aortic thrombus of patients who had symptoms of chronic peripheral arterial disease with limb-threatening ischemia (CLTI) (six cases) or typical distal embolism of “junk kick” type (two cases) (Figure 3). All CLTI patients were classified according to the Rutherford system as: class 4, one patient; class 5, four patients; and class 6, three patients. Two patients (25%) underwent one-step procedures, while six patients (75%) underwent hybrid one-step procedures. In two cases, coverage of the thoracic aorta embolic source was performed, while in six cases, the abdominal aorta was treated. The thoracic aorta was covered for an average of 130.5 mm (median 150 mm, IQR 150–101 mm). Five patients underwent femoro-tibial bypass and one underwent popliteal bypass. In five cases, they were performed using the large saphenous, while a prosthetic graft was used in one case. A total of eight patients (66.6%) underwent an aortic procedure after peripheral revascularization. All patients underwent postoperative antithrombotic and antiplatelet therapy (LMWH and acetylsalicylic acid), and were then discharged with antiplatelet therapy. Four patients (33.3%) underwent just the aortic procedure of the embolic source coverage and they were therefore discharged with a prescription for antiplatelet therapy (acetylsalicylic acid 100 mg per day) in combination with cilostazol for the treatment of PAD. No oral anticoagulation was administered because we performed an endovascular treatment for coverage of the embolic source. One embolic event was detected during procedures (with PSC2I). Transient renal failure occurred in one patient (8.3%), and there were no cerebrovascular events and no cases of spinal cord ischemia. In two cases, we observed a recurrent embolism after aortic coverage. Aortic-related mortality was reported in just one case of a patient who died of multiple organ failure Table 4. Overall survival was 100% at 30 days and 100% at 6 months. The overall primary patency of the distal bypass grafts was 100% at 30 days and 84% at 6 months in no PSC2I. The overall primary patency of distal bypass grafts in patients with PSC2I was 50% at 30 days and 47% at 6 months. Based on COVID’s onset timing, five patient subgroups were identified: one patient of eight submitted to hybrid procedures had a confirmed SARS-CoV-2 infection 1 month before intervention, three patients from 2 to 3 months before intervention, two patients from 5 to 6 months before intervention, one patient 8 months before intervention and the last patient 10 months before intervention.

## 4. Discussion

A relatively small number of publications, including case reports and small case series, describing patients with thrombus in a non-aneurysmal minimally atherosclerotic either thoracic or abdominal aorta have been published [10], while more recent experiences focus on the outcomes in conditions of an atherosclerotic and “shaggy” aorta [11]. Acute symptoms from peripheral embolism were the most frequent manifestation [6,7,8,10,12,13,14,15,16,17,18]. Diagnosis of this pathology is first based on the clinical evaluation of peripheral symptoms and signs, if present. Suspicions should be confirmed by instrumental diagnosis with the first step being the exclusion of any possible cardiological source. After the exclusion of arrhythmic causes, an echocardiographic examination should be performed in order to exclude anatomical anomalies and pathologies. Transesophageal echocardiography has an important role in the diagnosis of mural thrombus in aorta thoracic segments, especially in asymptomatic patients [19,20]. Its recent widespread diffusion permitted an improvement in the diagnostic process of this pathology. The CT angiography scan remains the gold standard for both the completion of the diagnostic process and the therapeutic strategy [3,4]. The optimal management of these patients is still controversial and it depends on the thrombus location and morphology, the symptoms and the patient’s general condition [10]. Treatment includes systemic anticoagulation and open surgical or endovascular procedures [15,20]. Surgical treatment has been rather restrictively used, mainly because operative morbidity is still considered comparatively high. The advancement of endovascular techniques and devices provides a minimally invasive alternative to patients otherwise deemed poor surgical candidates [8,21]. A limitation of this study is the lack of a histological confirmation of peripheral microembolism. Nevertheless, clinical evaluation is often sufficient for embolizing pathology diagnosis. Following this aspect, although there cannot be certainty about the treated aortic segment being the real embolizing source, the absence of recurrent embolizing events sustains the diagnosis ex juvantibus with a good probability. Shames and Carroccio reported results in their experiences that are similar to ours, with the latest having a rate of 89% of peripheral lesions healing after 1 year, in the absence of recurrent embolism [22,23]. Our data are in line with these, as we observed only two cases of recurrent embolism. One of these was probably due to an incomplete exclusion of the embolic source, which was confirmed by means of a CT scan angiography. This led to a revision of the previous lower limb bypass and a trans-metatarsal amputation. Instead, we noticed a difference in terms of renal failure after treatment, which was suffered in a transient form by three of our patients, all of them presenting a subsequent reversion to the values noted before treatment. We registered no cases of spinal cord ischemia. As it concerns the treatment of the thoracic aorta, in all our cases, the maximum length of coverage was 150 mm, seemingly following other literature about this being a prognostic factor in these terms. In total, 55% of our patients needed and underwent a lower limb revascularization procedure, with one of them needing a leg amputation after 1 month and one needing a leg amputation after 49 months. The improved accessibility to instrumental diagnostics resulted in better timing also related to an accidental diagnosis, resulting in an early treatment planification and therefore, probably, in a better result. Due to our center having a high volume of complex pathologies, it surely has an advantage regarding the diagnostic process. Still, embolic aorta remains a disease that is not well known as its treatment strategies remain controversial, especially in the context of a non-aneurysmal and minimally atherosclerotic aorta. Our experience underlines the importance of an early diagnosis in order to avoid complications on graft patency and for a patient’s general clinical conditions due to the persistence of an unrecognized embolic source. In the above-mentioned cases of patients who underwent the aortic procedure after peripheral intervention, the diagnosis of embolic aorta was made because of complications due to new embolic events, which caused the loss of patency of peripheral bypass or transient renal insufficiency and led to the execution of a CTA scan. It is important to perform antithrombotic therapy both in cases of conservative management and of surgical pathology treatment, as was suggested by many previous experiences. In addition, an antiplatelet therapy has to be performed after revascularization procedures and aortic endograft/stent coverage. Although not statistically significant, our data show a trend of better outcomes in favor of the group of patients who underwent synchronous aortic and peripheral procedures and the group undergoing aortic procedure first, among patients who underwent staged interventions. We registered two cases of recurrent embolism among patients who underwent the lower limb revascularization procedure before aortic coverage. In those patients, this led to a lower limb reintervention in order to maintain assisted patency. This shows a trend in favor of performing the aortic embolic source coverage first in terms of patency of lower limb bypass. The choice of different types of endografts seems not to have an influence in terms of outcomes, so this can be electively made on the base of anatomy and peculiar features of the aorta. The distance between the aortic and peripheral procedures did not seem to influence the outcomes directly, at least in the group of patients who underwent the aortic procedure first. However, the restricted number of patients could have a role in this observation. The multisystemic aspects of acute SARS-CoV-2 infection as major coagulation disorders and vascular complications such as thromboembolism have been thoroughly evaluated [24], while the long-term complications are still an unexplored area. SARS-CoV-2 invades vascular endothelial cells following a proinflammatory and procoagulant state. The hyperinflammatory response induces endothelitis [25], but it is not clear how long this condition persists in the disease convalescent phase. The risk of vascular complications in post-COVID-19 infection is probably due to its hyperinflammatory condition. The late onset of thrombotic events after COVID-19 infection linked to a hypercoagulable state with no preexisting occurrence before COVID-19 infection has been described [26]. From our experience, we described an increased mortality and amputation rate in patients with previously confirmed SARS-CoV-2 infection submitted to intervention for mural aortic thrombosis associated with the chronic peripheral obstruction of lower limbs. In these patients, persistent hematologic (as coagulopathy and elevated base line D-dimers) and immunologic disease can persist after acute COVID-19 [27].

### Limitations

The main limitations of our study include a relatively small number and heterogeneity of patients.

## 5. Conclusions

Our experience with stent coverage of complex aortic lesions, which could be performed alone or in association with distal bypass graft for embolization-induced PAD, supports this approach to this pathology in a variety of settings. The COVID-19 pandemic has led to an increased prevalence of mortality and amputation in patients with mural aortic thrombosis associated with chronic peripheral obstruction of the lower limbs due to the interconnection between risk factors such as corticosteroid therapy, hematologic disease and procoagulant and proinflammatory pathways following SARS-CoV-2 infection.

## Figures and Tables

**Figure 1 diagnostics-13-01208-f001:**
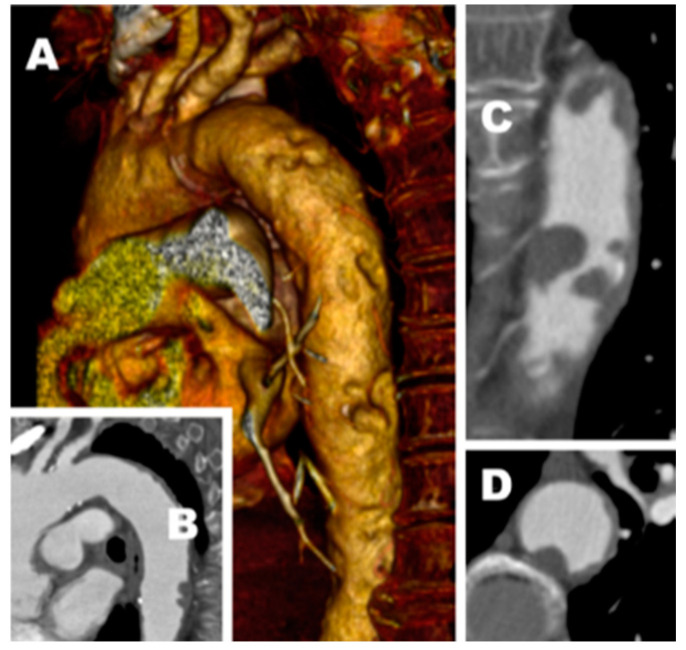
CTA scan revealed (**A**) multiple focal ulcerative embolic plaques, (**B**) minimal injury lesion, (**C**) irregular ulcerative plaque, (**D**) excavated lesion.

**Figure 2 diagnostics-13-01208-f002:**
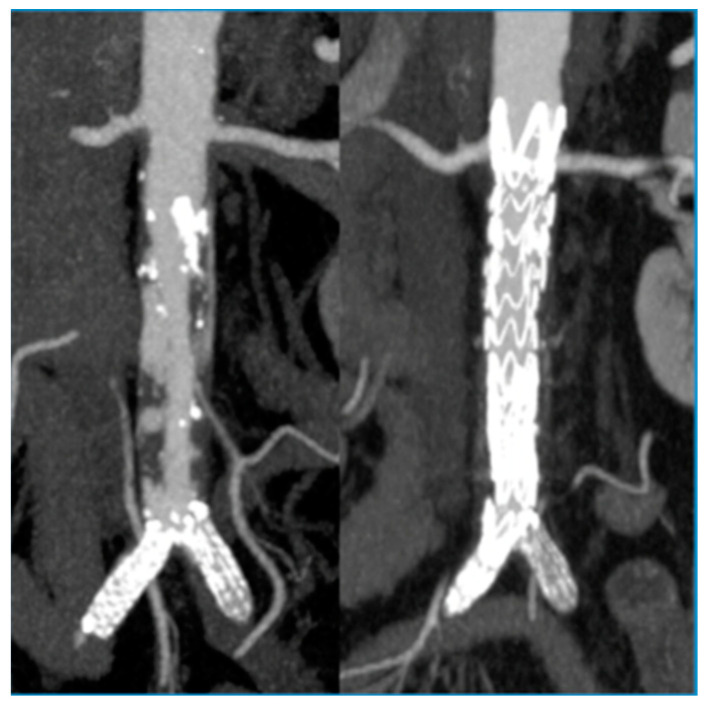
Abdominal aortic endoprosthesis.

**Figure 3 diagnostics-13-01208-f003:**
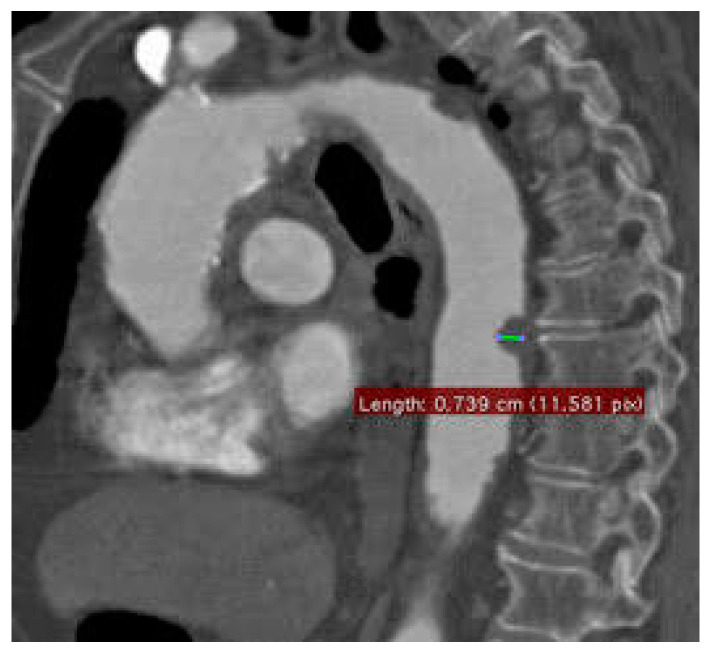
CTA scan revealed unstable thoracic aortic thrombus.

**Table 1 diagnostics-13-01208-t001:** Characteristics of patients.

		Patients: *n*. (%)
Age	68.8 ± 10.2	
Male		41 (79%)
Hypertension		44 (81.5%)
Diabetes		22 (40%)
Smoking habit (current)		30 (55%)
Chronic Obstructive Pulmonary Disease		22 (40%)
Chronic Kidney Disease		12 (22%)
Coronary Artery Disease		17 (31%)
Antiplatelet drugs		30 (55%)
Antihypertensive drugs		43 (79%)
Chronic limb-threatening ischemia symptoms and signs		39 (72%)
Acute limb ischemia symptoms and signs		24 (54%)

**Table 2 diagnostics-13-01208-t002:** Operative details of patients.

	Patients: *n*. (%)
Thoracic aortic source	19 (35%)
Abdominal aortic source	35 (64%)
Hybrid treatment	30 (55%)
- Synchronous	12 (22%)
- Staged	22 (40%)
- Aortic procedure first	8 (14%)
- Peripheral procedure first	11 (20%)
Aortic procedure only	24 (44%)

**Table 3 diagnostics-13-01208-t003:** Operative details: used endografts patients.

	Patients: *n*. (%)				
	Bolton Relay/Relay NBS+	Cook Zenith Alpha	Medtronic Valiant Navion/Endurant II	Gore TAG/Excluder	Endologix AFX
Thoracic aortic source	10 (18%)	1 (2%)	4 (9.5%)	4 (9.5%)	/
Abdominal aortic source	/	8 (14%)	12 (22%)	5 (9%)	8 (14%)

**Table 4 diagnostics-13-01208-t004:** Outcomes.

	Patients: *n*. 54 (%)
Transient renal failure	4 (7.4%)
Amputations	13 (24%)
Fasciotomy	5 (9.2%)
Reperfusion syndromes	4 (7.4%)
Recurrent embolism	3 (5.5%)
Embolic stroke	1 (1.8%)
Multiple organ failure (MOF)	1 (0.8%)

## Data Availability

The data that support the findings of this study are available from the corresponding author (F.B. and G.D.C.) upon reasonable request.

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
