# Peer review of "Single-Center Experience with Simultaneous Mural Aortic Thrombosis and Peripheral Obstructive Disease in Pre-COVID-19 and COVID-19 Era"

_diagnostics, 2023, doi:10.3390/diagnostics13061208_

Round 1

Reviewer 1 Report

Thank you for your submission. Congrats on your study!

Abstract:
- First sentence slightly ambiguous. Are you referring to patients with acute on chronic limb ischaemia, precipitated by acute emboli from aortic mural thrombus?
- Line 16/17: single-centre experience
- Line 30: aortic-related mortality
- Conclusion: Not clear. Grammar needs to be improved

Intro:
- It is important to highlight your study for patients with acute on chronic limb ischaemia, precipitated by acute emboli from aortic mural thrombus.

Methods:
- These patients have acute on chronic limb ischaemia. They should be presented with Rutherford's classification for acute limb ischaemia, and not for chronic limb ischaemia
- How were patients selected for open vs endovascular procedures?
- You'd mentioned post-op anti-coagulation with LMWH. Are the patients discharged on oral anti-coagulation (warfarin / rivaroxaban / apixiban / etc)?

Results:
- Line 137: lost to follow-up
- Please provide data/table on the outcomes (acute kidney injury, reperfusion syndromes, fasciotomies, amputations, etc)

Discussions:
- Please divide into paragraphs
- Please provide a paragraph on your study limitations

Author Response

Point 1:

Abstract:
- First sentence slightly ambiguous. Are you referring to patients with acute on chronic limb ischaemia, precipitated by acute emboli from aortic mural thrombus?

Response 1: We' ve added :" Repeated embolism of instability aortic mural thrombosis caused acute limb ischemia (Rutherford 2 Classification) in patients with Peripheral Arterial Disease (PAD)"

Point 2: - Line 16/17: single-centre experience

Response 2:  We've corrected this

Point 3:  Line 30: aortic-related mortality

Response 3: We've corrected this

Point 4:  Conclusion: Not clear. Grammar needs to be improved

Response 4: We've corrected the conclusions

Point 5: Intro:
- It is important to highlight your study for patients with acute on chronic limb ischaemia, precipitated by acute emboli from aortic mural thrombus.

Response 5: We’ ve highlighted as a chronic limb ischemia precipitated in acute limb ischemia by aortic embolism 

Point 6: Methods:
- These patients have acute on chronic limb ischaemia. They should be presented with Rutherford's classification for acute limb ischaemia, and not for chronic limb ischaemia

Response 6: We’ ve specified the patients in Rutheroford calssification for PAD e Rutherford classification for ALI

Point 7: How were patients selected for open vs endovascular procedures?

Response 7: Decision making for open vs endovascular procedures based on patients' comorbidity

Point 8: You'd mentioned post-op anti-coagulation with LMWH. Are the patients discharged on oral anti-coagulation (warfarin / rivaroxaban / apixiban / etc)?

Response 8: The patients has been discharged  on antiplatelet therapy (acetylsalicylic acid 100 mg per day) in combination with cilostazol for treatment of PAD. . No oral anticoagulation was administered because we performed an endovascular treatment for the coverage of embolic source.

Point 9: Results:
- Line 137: lost to follow-up

Response 9: We've corrected this

Point 10: - Please provide data/table on the outcomes (acute kidney injury, reperfusion syndromes, fasciotomies, amputations, etc)

Response 10: We add a data/table on the outcomes (Table

Point 11: Discussions:
- Please divide into paragraphs

- Please provide a paragraph on your study limitations

Response 11: We've added study limitation

Reviewer 2 Report

Dear authors,

I find your paper interesting. Anyway, I do not agree with your comparison between group 1 and 2: the sample sizes are too different, I think thant you should write about mural thrombosis in a unique group and you could identify Covid-19 infection as one of the main risk factors for amputation/mortality. It is a major revision required, but once provided, the paper can be published. 

Author Response

Point 1: I do not agree with your comparison between group 1 and 2: the sample sizes are too different, I think thant you should write about mural thrombosis in a unique group and you could identify Covid-19 infection as one of the main risk factors for amputation/mortality. It is a major revision required, but once provided, the paper can be published. 

Response 1: We've identified a unique group as you suggest

Reviewer 3 Report

maybe more recent references would be useful

Author Response

Point 1:

maybe more recent references would be useful

Response 1:  We' ve added more recent reference " Journal of Vascular Surgery Cases, Innovations and Techniques Volume 6, Issue 3, September 2020, Pages 483-486
COVID-19 and vascular disease Acute aortic thrombosis in COVID-19

Round 2

Reviewer 2 Report

Dear Authors,

I find your paper better, as you identified only one group. Good job!